# Telemedicine in Hematopoietic Cell Transplantation and Chimeric Antigen Receptor-T Cell Therapy

**DOI:** 10.3390/cancers15164108

**Published:** 2023-08-15

**Authors:** Arpita P. Gandhi, Catherine J. Lee

**Affiliations:** 1Center for Hematologic Malignancies, Knight Cancer Institute, Oregon Health & Science University, Portland, OR 97239, USA; 2Fred Hutch Cancer Research Center, Clinical Research Division, Seattle, WA 98109, USA

**Keywords:** hematopoietic stem cell transplantation, CAR-T, telemedicine, palliative care, wearable devices, COVID-19, PHE, public health emergency, hematologic malignancies, chimeric antigen receptor-T cell, chronic care model

## Abstract

**Simple Summary:**

Patients with high-risk acute leukemias and lymphomas undergo stem cell transplantation or cellular therapies to achieve a cure. These are complex treatments that are typically offered by specialty medical centers. Patients are treated and monitored in the hospital for 3–4 weeks due to the potential for life-threatening complications. While it is still in its infancy, the use of telemedicine is changing this treatment paradigm by allowing patients to be in the comfort of their own home. We review data on how telemedicine is being incorporated for patients who receive stem cell transplantation or cellular therapy.

**Abstract:**

Telemedicine has played an important role in delivering healthcare for primary care, chronic disease patients, and those with solid organ malignancies. However, its application in subspecialties such as hematologic malignancies, hematopoietic cell transplantation (HCT), or chimeric antigen receptor-T cell (CAR-T) therapy is not widespread since physical examination is a vital component in delivering care. During the COVID-19 pandemic, we widely used telemedicine, since protecting our immunocompromised patients became our top priority. The employment of HCT and CAR-T therapies continues to grow for high-risk hematologic malignancies, particularly in older and frail patients who must visit specialty centers for treatment access. Generally, HCT and CAR-T therapy care is highly complex, necessitating commitment from patients, caregivers, and a multidisciplinary team at specialty academic centers. All healthcare systems adapted to the crisis and implemented rapid changes during the COVID-19 public health emergency (PHE). Telemedicine, a vital modality for delivering healthcare in underserved areas, experienced rapid expansion, regardless of the geographic region, during the COVID-19 PHE. The data emerging from practices implemented during the PHE are propelling the field of telemedicine forward, particularly for specialties with complex medical treatments such as HCT and CAR-T therapy. In this review, we examine the current data on telemedicine in HCT and cellular therapy care models for the acute and long-term care of our patients.

## 1. Introduction

The 21st century has seen a surge in telemedicine utilization, with the COVID-19 pandemic further accelerating its adoption due to restricted in-person visits and the critical need to minimize exposure for immunocompromised patients. Patients with autoimmune illnesses requiring immunosuppression, cancer diagnoses undergoing anti-cancer treatments, or those who received solid organ transplantation, HCT, or CAR-T therapy faced heightened risks for adverse outcomes with the COVID-19 illness [1,2]. This scenario prompted rapid practice changes at academic centers and community practices worldwide. In HCT and cellular therapy, physicians had to quickly adapt and integrate telemedicine into the care of patients as both therapies demand frequent health assessments by multidisciplinary providers during that acute care period, particularly in the initial 100 days after treatment. The care of long-term survivors of allogeneic (allo) HCT with complications, such as chronic graft-versus-host disease (cGVHD), also required a transition to telemedicine, as their immunosuppressive treatments put them at high risk for COVID-19 infection and illness.

The lessons learned during the PHE have been invaluable for HCT and CAR-T patients. Future studies will determine the role of telemedicine in providing patient-centered care while overcoming geographical and financial barriers.

## 2. Telemedicine in the Cancer Care Continuum

Telehealth and telemedicine, although used interchangeably, have distinct meanings. Telehealth refers to a broader application that supports long-distance clinical healthcare, patient and professional health-related education, public health, and health administration using electronic information and telecommunication technologies. Telemedicine, on the other hand, involves the practice of medicine using electronic communication, information technology, or other means between a physician in one location and a patient in another, with or without an intervening healthcare provider [3]. Telemedicine interventions utilize various platforms, such as videoconferencing, electronic medical record (EMR) platforms, third-party applications, hub-and-spoke facilities, and smartphone technologies, to provide a safe and accessible platform for healthcare, especially in rural areas where access is limited.

There are four main modalities of telemedicine: (1) synchronous (real-time) communication, (2) asynchronous (store-and-forward) sharing of medical data, (3) remote patient monitoring, and (4) mobile health (mHealth) via wearable devices and smartphone apps [4]. The synchronous modality of telemedicine gained popularity during the COVID-19 pandemic, allowing patients, caregivers, and healthcare providers to communicate in real-time and without the need for masks. Telemedicine can facilitate patient care in all stages of the cancer care continuum. It is an effective means for visits that generally do not require a physical exam, such as for cancer screening and genetic counseling. For example, in a randomized trial (TeleCARE), individuals at risk for colon cancer were randomized to a telehealth-based, personalized intervention that incorporated evidence-based risk communication and behavior change techniques vs. a mailed educational brochure. In the intent-to-treat analysis, the telehealth group (35.4% vs. 15.7%) was almost three times more likely to be screened with a colonoscopy (odds ratio, 2.83; 95% CI, 1.87–4.28; *p* < 0.001) [5]. The National Society of Genetic Counselors (NSGC) is also incorporating telehealth modalities in practice and recently published their first practice guidelines for genetic counselors [6,7].

Patients with hematologic malignancies often face travel and financial challenges when receiving HCT or CAR-T therapy at large academic centers. In the United States, patients are typically required to relocate for up to 100 days with caregivers providing 24 h support. During this time, patients are seen approximately twice a week to monitor for acute graft vs. host disease (aGVHD), cytokine release syndrome (CRS), immune effector cell-associated neurotoxicity syndrome (ICANS), resolution of chemotoxicities, and supportive care with transfusions and intravenous fluids (IVF). Historically, telemedicine has seen minimal use in managing the recipients of HCT or CAR-T therapy. However, small pilot and retrospective studies have demonstrated the feasibility of incorporating telemedicine in these settings, and future trials will explore its use in highly complex treatments with cellular therapies.

## 3. Telemedicine in Hematopoietic Cell Transplantation

### 3.1. Telemedicine and Access to Care

Telemedicine has emerged as a valuable tool in HCT and cellular therapy, offering the potential for increased accessibility to specialized care and to alleviate time and financial burdens associated with in-person visits. In a study on geographic access to HCT services in the United States (US) in 2015, there were 229 facilities that provided services for approximately 306 million people, which is an average of 1 HCT facility per 1.3 million people in the US [8]. Of the US population, 46% resided within 30 min of an age-appropriate HCT center, and access to HCT care was limited for people residing outside of that zone. It meant that 30–50% of the US population would have to travel a total of 120–180 min to receive care at an HCT center. In a study conducted at an alloHCT center in Brazil, the majority of patients (*n* = 232) had access to smartphones or personal computers, enabling them to participate in a survey evaluating the time and financial burdens associated with in-person clinic visits [9]. The results revealed that 33% of patients spent over 120 min traveling to and from the clinic, while 42% incurred commuting costs exceeding US dollar (USD) 10.00. Additionally, 38% of participants experienced some degree of debility, and 28% complained about long waiting times for in-person visits with their physician. Unfortunately, travel time combined with wait time in the office and time during the office visit create time and financial toxicity for patients. As a result, patients may struggle to adhere to follow-up appointments. Telemedicine plays a role in improving access to HCT services as well as reducing time and financial toxicities, particularly for patients who are considered clinically stable after HCT.

### 3.2. Telemedicine in Outpatient or Homebound HCT

Small pilot studies and outcomes with institutional practices have reported feasibility on the use of telemedicine in HCT patients and that it may complement the homebound HCT models. For example, at Duke University in Durham, North Carolina, they conducted the first homebound transplant study in the US (2012–2018), which incorporated telemedicine for autologous (auto) HCTs (*n* = 17) and alloHCT (*n* = 8) recipients (Table 1) [10]. Patients were required to live within 90 min from the transplant center and have 24 h caregivers available. In the experimental arm, patients (*n* = 25) received a conditioning regimen and cell infusion in the hospital or in the outpatient clinic and were discharged to home on day +1. Subsequently, an advance practice provider (APP) conducted a daily home visit for a physical exam and blood draw, followed by a videoconference with the attending physician. Compared to the SOC outpatient or hospital-based HCTs, there was no significant difference in the outcomes (relapse, mortality, and GVHD at 1 year) of the experimental arm with telemedicine. Quality of life (QoL), as assessed by the validated Functional Assessment of Cancer Therapy-Bone Marrow Transplant (FACT-BMT), for the home HCT group was significantly higher at day 100 in homebound autoHCTs compared to the matched controls. While this pilot study was conducted in the pre-COVID-19 era, these experiences and expertise became valuable during the pandemic when in-person interactions had to be minimized. The Duke University program swiftly changed their practice and conducted frequent house calls by a BMT nurse along with a video visit with the physician [11]. A similar study at Memorial Sloan Kettering Cancer Center (MSKCC) showed the feasibility as well as patient and caregiver preference for a homebound HCT model supported by telemedicine [12]. However, 80% of patients (12 of 15) experienced issues with the technology used for telemedicine, with Wi-Fi connectivity being a common problem. The quality of the Wi-Fi connection has since improved and is less likely to be a true barrier for this model of care. 

### 3.3. Telemedicine in the Remote Monitoring of HCT

The incorporation of mobile health (mHealth) in healthcare has been transformative since the early 2000s, providing novel methods for remote patient monitoring, point-of-care diagnostics, and medication management without needing in-person clinic visits. It also represents an opportunity to reduce caregiver burden since this method can provide direct access to patient data for the care team’s evaluation. In a Spanish study called the SMARTCOVID19 study, allo/autoHCT patients (*n* = 16) and their caregivers participated in an outpatient monitoring plan for 14 days [13]. A smartphone application was used to collect data on vital signs and symptoms for 14 days after hospital discharge. In this feasibility study, an alarm was programmed for preset parameters. If activated, a patient would have a telemedicine visit to evaluate the symptoms, followed by an in-person visit. Of the 16 patients, 4 did not complete the study due to their inability to work with the smartphone application, while the 12 patients who completed the study reported adherence and favored this form of monitoring. Clinicians were able to diagnose new-onset hypertension and aGVHD of the skin with this remote monitoring strategy.

The group at Duke University conducted the first pilot study for alloHCT patients using mHealth modality and an activity tracker. They employed a mobile application (app) named TRU-BMT (Technology Recordings for Better Understanding of Bone Marrow Transplantation) for the early detection of adverse events with goals to implement early interventions for improved outcomes [14]. This app allowed patients to enter information such as their general health, food intake, stool count, exercise and sleep habits, levels of stress, and other symptoms. Patients could also receive a report showing longitudinal trends that would help them understand their progress over time. Patients werealso given activity trackers (Apple Watch or Microsoft Band) to monitor real-time physiological data, such as heart rate and step counts up to 180 days post-alloHCT. At the completion of the pilot study, the TRU-BMT application adherence (primary endpoint) was 30% for daily use and 43% for weekly use, and the activity tracker adherence (secondary endpoint) was 58% for daily use and 83% for weekly use. The overall adherence was comparable to other mHealth applications. TRU-BMT’s symptom severity (SS) was associated with HCT outcomes, wherein a lower SS was significantly associated with a shorter hospital stay (OR: 0.90; 95% CI: 0.82–0.99; *p* = 0.036) and decreased odds of moderate or higher cGVHD (Odds Ratio (OR): 0.24; 95% CI: 0.06–0.87; *p* = 0.003). The phase 2 trial of TRU-BMT evaluated HCT outcomes in patients randomized to TRU-BMT vs. SOC. This tool was also customized for pediatric patients, and TRU-PBMT and its applications for this population are eagerly anticipated [18]. Remote monitoring using novel approaches such as the mobile app and wearable devices has the potential to improve the scope of telemedicine and provide means for the early diagnosis of new complications without increasing the patient or caregiver burden. 

## 4. Telemedicine for Long-Term Care after alloHCT

### 4.1. Survivorship after alloHCT

The cumulative incidence of chronic health conditions in alloHCT survivors is 64% at 10 years and approaches 71% at 15 years [19]. The burden of chronic conditions results in reduced survival, poor QoL, and increased financial burden. It is expected that by the year 2030, the number of HCT survivors will increase five-fold, resulting in half a million survivors in the US alone [20]. Patients typically return to their home and reestablish with their primary oncologist after 100 days of HCT care. During this time, patients require monitoring for signs of cGVHD, disease relapse, new comorbidities, and immunizations. Early recognition of cGVHD and disease relapse is critical for timely diagnostic work-up and treatment which can lead to improved outcomes [21,22]. Some patients are unable to attend an in-person visit due to long travel distances to the HCT center, financial hardships, work or school commitments, or the inability to secure a caregiver for the family. There is a trend towards the discontinuation of long-term care, especially in the adolescent and young adult (AYA) population [23]. A recent systematic review showed that there was substantial evidence that telemedicine was beneficial in the management of psychosocial and physical effects, particularly for improving fatigue and cognitive function in cancer survivors [24]. Telemedicine may be able to bridge this gap, particularly for patients who are unable to attend in-person visits at the transplant center. 

The long-term follow-up (LTFU) program at Fred Hutchinson Cancer Research Center (FHCRC) is a well-established telemedicine program for alloHCT survivors that was designed to support community oncologists and primary care physicians in managing post-alloHCT complications and survivorship care (Table 2) [25]. The program receives over 3,200 queries annually from patients and community providers, with most queries relating to diagnosing and managing cGVHD, infections, immunizations, disease relapse, subsequent primary malignancies (SPMs), and psychosocial issues. Photographs, laboratory, or imaging findings are reviewed at daily telehealth rounds and contribute to the overall improved outcomes of long-term survivors at FHCRC. This program runs on institutional funds, and the telemedicine services are not billed to the patient. While this is a successful model serving its community in the State of Washington, it is not feasible for all transplant centers to have a similar model since it is resource-intensive and requires a substantial budget to start this type of program. 

In a study at the City of Hope Cancer Center, HCT survivors at one year were remotely monitored for one month if they were at risk of cardiovascular disease [26]. Patients received an mHealth kit, and data were monitored by a registered nurse (RN) with expertise in interpreting and validating remote patient-reported data. The RN conducted follow-up calls for patients with data alerts, triggered calls to physicians for further instructions, or sent patients to the emergency department if high-risk patterns were detected. While this was a short one-month feasibility study, increased engagement and compliance were reported after the first week, and further refinement on barriers can increase its use in monitoring other key components of survivorship care. 

### 4.2. Chronic GVHD after alloHCT

cGVHD is one of the main complications in long-term survivors, occurring in about 40–50% of alloHCT survivors. It can present with mild to severe manifestations, involving any organ in the body. Patients with mild or early manifestations of cGVHD have better outcomes compared to those with severe forms, which are generally associated with disability and poor QoL. Early detection of cGVHD manifestations, such as bronchiolitis obliterans syndrome (BOS), is critical for timely diagnosis and interventions to improve outcomes in long-term survivors. In patients who undergo lung transplantation, it is a routine practice to monitor for acute or chronic lung rejection using home spirometry testing. In a feasibility study at the FHCRC, patients at risk for developing BOS underwent remote monitoring with a home spirometer [27]. Data were wirelessly transmitted to a cloud-based monitoring portal and 78% of patients completed one year of weekly monitoring [27]. There was a high correlation between the home FEV1 and laboratory FEV1 values. Twelve patients were diagnosed with BOS or suspected BOS, and nine had antecedent FEV1 decline detected by home spirometry. The use of home spirometry for remote monitoring can potentially lead to the early detection of BOS in alloHCT survivors and improved outcomes. Other manifestations of cGVHD, such as involvement of the eyes, oral cavity, skin, and joints do require an in-person examination by a specialty physician for accurate diagnosis. Chronic GVHD is a heterogenous disease, and these patients will benefit from a combined model that allows for virtual connection with the transplant center when new issues arise, followed by an in-person examination if indicated. An Italian group reported their use of telemedicine during the COVID-19 PHE, which could be a potential model for future studies in long-term care after alloHCT. They called adult HCT survivors, using predetermined decision-making criteria to identify patients who were appropriate for an in-person visit or telemedicine visit [28]. If COVID-19 was suspected or detected, these patients were managed by a dedicated COVID-19 team. Patients who did not have COVID-19 or acute issues were seen via telemedicine. Alternatively, patients who reported HCT-related complications were seen in the clinic. They called 236 (51%) of 465 adult alloHCT patients, and 40% of the long-term survivors were on immunosuppressive therapy (IST) for cGVHD, placing them at high risk for infections. Using their predetermined criteria, they conducted telemedicine visits in >30% of these immunocompromised patients and the rest were seen at their LTFU clinic. Transplant physicians felt confident in their risk-stratified decision-making criteria, and they plan to incorporate this strategy into their post-pandemic practice. In Italy, the national system and COVID-19 practices during the PHE did not allow a discussion on the legal and financial framework of this kind of program. Their data suggest it is feasible to incorporate telemedicine into their LTFU program.

In conclusion, the integration of telemedicine in the care of alloHCT survivors is feasible and has shown promising results in the early detection of complications, patient adherence, and overall satisfaction. Telemedicine has the potential to bridge the gap in care for patients who may have difficulty attending in-person visits, offering better access to specialized medical providers and information. Although several feasibility studies have demonstrated the benefits and reliability of telemedicine in this population, further research on a larger scale is needed to refine these models and establish standardized practices that can be incorporated into the overall care of alloHCT survivors.

## 5. Telemedicine for Supportive Care

### 5.1. Psychological Support

Chronic health conditions are common in HCT survivors, with 45% of autoHCT and 70% of alloHCT patients having at least one chronic health condition at 5 years [33]. Many survivors also experience emotional distress and depression due to the burden from chronic health conditions and isolation after alloHCT, predominantly due to their immunocompromised state requiring them to isolate themselves from large crowds and potentially sick family members. The INSPIRE randomized controlled trial (RCT) was designed to assess the efficacy of a personalized Internet-based survivorship care program targeting emotional distress, depression, and fatigue compared with a control group [29]. The Internet site contained five major sections: boosting health; restoring energy; renewing outlook; getting connected; and tips and tools addressing other topics such as sleep, sexual function, and memory. After randomization to this arm, participants were sent a welcome e-mail with the INSPIRE site, where they were greeted by a personalized welcome page and suggestions for pages to visit as determined by patient-reported outcome scores. The participants had the following characteristics: the mean age of participants was 51 years; predominantly Caucasian (93.8%); urban residence (79%); had >2 years of college education (72%); annual income of USD 40,000–USD 80,000 (28%) and >USD 80,000 (53%); and the majority had an intermediate/expert level of computer experience (86%) [30]. The primary analysis of this RCT demonstrated that those who used the INSPIRE online program with its companion problem-solving telehealth calls were significantly more likely to improve from the “distressed” to “not distressed” status. They described patterns of engagement and demonstrated that engagement was important for improved outcomes. The factors associated with increased engagement included age ≥40 years, female sex, active cGVHD, and higher distress and fatigue scores [30]. In this study, African Americans were less likely (*p* < 0.001) to enroll while native American/Alaska natives were more likely (*p* = 0.03) to enroll in the study when compared to white survivors post-HCT [31]. This highlights the need to address potential barriers to access and ensure that telemedicine programs are inclusive and accessible to a diverse patient population. In conclusion, the INSPIRE trial demonstrated the potential benefits of personalized Internet-based survivorship care programs in improving emotional well-being and reducing distress among HCT survivors. Further development and refinement of such programs, along with efforts to increase accessibility and inclusivity, can help enhance the QoL for HCT survivors and better address their chronic health and emotional needs.

### 5.2. Physical Activity and Frailty

Frailty and physical deconditioning are common issues despite significant advancements in the field of HCT and are known to increase the risk of mortality. Exercise has been an established strategy to overcome this complication. Traditionally, exercise programs for cancer patients are conducted through a rehabilitation center or a physical therapy center. Lee, K., et al. conducted a feasibility study of implementing an 8-week-long supervised exercise intervention in pre-frail or frail survivors of HCT [31]. Patients were randomized to telehealth exercise vs. control, and a telehealth exercise intervention using exercise resistance bands was conducted over 30–60 min per session, three times per week. Of the 20 patients, 18 were classified as frail and 9 had cGVHD at the time of enrollment. In the feasibility study, there was more than 90% of the participants who completed >70% of the prescribed exercise sessions. While the numbers were small in this study, the authors did report improved gait speed and increased hand grip in the exercise group. Frail patients would have struggled to attend an in-person exercise program three times per week, and this strategy can help break the access to care barrier for already disadvantaged patients. The integration of telemedicine for exercise intervention will help to reduce debility and healthcare utilization by these patients. 

### 5.3. Palliative and Hospice Care

Palliative and hospice care services play a crucial role in the cancer care continuum, offering support and symptom management to patients and their families [34]. The incorporation of telemedicine in palliative care services is reported in primary cancer treatment. It could potentially address the overwhelming number of appointments experienced by patients after HCT, particularly during the critical 100 days after HCT. However, studies on the effectiveness of telemedicine in palliative care have shown mixed results. A randomized trial in advanced cancer patients reported increased anxiety in the telemedicine group compared to the standard approach in palliative care [35]. Nonetheless, telemedicine was adopted in cancer and HCT patients during the PHE, leading to some positive outcomes. An observational study from Taiwan used smartphone-based family conferences to discuss end-of-life care for patients [32]. While the majority of the participants were neutral or satisfied with the experience, the study revealed some challenges. Some family members found it convenient to join the meetings remotely, but others encountered issues with small screens, connectivity problems, or a preference for face-to-face conversations. In conclusion, telemedicine has the potential to support palliative care services for cancer and HCT patients, offering convenience and accessibility. However, it is essential to address the technical challenges and preferences of patients and their families in order to optimize the effectiveness of this approach. Further research and the development of tailored telemedicine programs could help to improve patient and/or family satisfaction and enhance the delivery of palliative care services in this setting [36].

## 6. Barriers with Telemedicine

### 6.1. Reimbursement

Reimbursement for telemedicine visits varies across the world and it has historically been a significant barrier to its widespread adoption in the United States. Telephone visits are reimbursed at lower rates compared to video and in-person visits, which can disincentivize physicians and institutions from providing care via telemedicine, particularly for complex patients who receive HCT and CAR-T therapy. However, the COVID-19 pandemic prompted changes in reimbursement and licensure requirements, leading to a rapid expansion of telemedicine services in the US. Before the PHE, reimbursement for telemedicine services was limited to certain medical specialties and predominantly focused on rural populations. Since 6 March 2020, Medicare expanded reimbursements for telemedicine services to all originating locations, including patients’ homes. The waiver also allowed for the delivery of telemedicine services across state lines and waived the requirement for providers to hold licensure in the state of the originating site at the time of the telemedicine service. These changes resulted in a significant increase in telemedicine utilization. Telemedicine visits accounted for 41% of all primary care visits in April 2020, up from just 0.1% in January 2020 [37]. The total number of telemedicine visits also saw a substantial increase, rising from approximately 840,000 in 2019 to nearly 52.7 million in 2020 [38]. It is crucial to continue monitoring reimbursement policies and licensure requirements to ensure that they promote the safe and effective delivery of telemedicine services, especially for complex patients. 

### 6.2. Access to Internet

The success of telemedicine is based on successful Internet access and connectivity. For example, in a study at an urban transplant center in New York City, patients and physicians preferred telemedicine visits over in-person visits after alloHCT [15]. However, there were connectivity issues, and they did not obtain high resolution images for accurate GVHD grading and assessments. Despite these issues, the study found that telemedicine visits were able to capture moderate to severe symptoms using the M D Anderson Symptom Inventory surveys and were particularly useful in patients who were clinically well and therefore the visit focused on symptom checks or medication clarifications. 

Other barriers that impact access to the Internet and telemedicine include age, language, and socioeconomic status. Nearly all studies evaluating telemedicine will exclude patients who do not speak English or the native language of the respective country. During the COVID-19 pandemic, there was a successful incorporation of virtual interpreters on smart or electronic tablets, which were used during inpatient rounds and virtual visits. We need studies that will formally evaluate the use of virtual interpreters in telemedicine for post-alloHCT patients. 

Historically, older patients have been digitally disadvantaged due to a lack of access to or experience with virtual technology. Other limitations have included vision, hearing, or communication impairments or cognitive decline that impede virtual communication. A considerable increase in the use of the Internet occurred in 2016 with two-thirds of adults over the age of 65 reporting Internet use compared to 12% in 2000; however, this was considerably lower compared to the 90% adoption rate of the Internet for the overall population. Furthermore, the adoption of faster broadband Internet at home, via a smartphone or desktop, also lagged in the older population at 40–50% in 2016, thereby creating barriers to the successful use of telemedicine services for this patient population [39]. However, some older patients may benefit from telemedicine for routine follow-up, particularly if they are limited by physical function and struggle to travel for an in-person visit. Future studies in older patients undergoing HCT should evaluate telemedicine as a means to increase access to care and potentially improve QoL for older patients and their caregivers. 

## 7. Integrated Healthcare Model for alloHCT

Currently, there is not a well-established care model that is comprehensive and focuses on the unique needs and challenges of HCT patients in the real world. Patients are alloHCT may gain the maximum benefit from telemedicine when it will beused in a specialized chronic care model (CCM), since patients’ needs are likely to change based on active complications. The CCM is a care model that was designed to address essential elements of chronic disease, where the goal is to improve outcomes by directly addressing the needs of a chronically ill population and to also guide physicians to create a practice that is highly patient-centered. It has four dimensions, including self-management support, decision support, clinical information systems, and delivery system designs [40]. The updated eHealth enhanced CCM (eCCM) outlines how to strengthen all four dimensions via digitalization. Incorporating eHealth-facilitated integrated care models tends to improve biomedical, behavioral, psycho-social, and economic outcomes in patients with chronic illnesses such as diabetes, rheumatologic illnesses, and solid organ transplantation. Medication compliance and reduced re-hospitalizations have been reported in solid organ transplant recipients [41,42]. In cancer care, these integrated care models (ICM) have improved the management of the patients’ symptom burden, rate of re-hospitalizations, survival, QoL, and physical activity.

The SMILe-ICM program (SteM-cell-transplatatatIon faciLitated by eHealth), actively enrolling in Switzerland, represents a novel approach to integrated, eHealth-enabled care for patients undergoing alloHCT [43]. The program is comprised of four self-management intervention modules: targeted monitoring and follow-up of key medical and symptom-related parameters, infection prevention, medication adherence, and physical activity. Phase 1 of the SMILe-ICM’s development involved outlining its theoretical foundation through surveys of patients and clinicians. These surveys revealed that the existing care model was primarily driven by acute care, conducted by inpatient and outpatient teams, with limited collaboration between care teams, and was not patient-centered and did not include self-management strategies [44]. During phase 2–4 of this program development, SMILe intervention modules and delivery methods were identified, using patient-centered approaches [43,44,45,46,47]. As the program continues to evolve, it will be essential to assess the impact of eHealth interventions on patient outcomes and healthcare delivery. In their randomized clinical trial (NCT04789863), 80 consecutive adults undergoing alloHCT will be enrolled at the University Hospital of Basel in Switzerland and undergo 1:1 randomization into a SOC group or SMILe-ICM group. These interventions will be delivered either face-to-face or via the SMILeApp that allows patients to connect with their transplant team and allows the team to monitor and rapidly intervene in patient-reported parameters within the first year of HCT. The primary outcome of this trial is the rate of re-hospitalization, and the secondary outcomes include healthcare utilization costs, length of inpatient re-hospitalizations, medication adherence, treatment and self-management burden, QoL, GVHD, and overall survival, as well as implementation outcomes such as acceptability, appropriateness, feasibility, and fidelity. If successful, this model may be the future for the delivery of care with alloHCT patients.

## 8. Telemedicine in Chimeric Antigen Receptor-T (CAR-T) Cell Therapies

### 8.1. Outpatient Programs for CAR-T Cell Therapy

CAR-T cell therapies have become a critical part of the treatment algorithm in patients with B-cell lymphomas, acute lymphoblastic leukemia, and multiple myeloma. Over the last decade, centers have gained expertise in managing and monitoring its toxicities, particularly CRS and ICANS. Outpatient CAR-T cell infusion and monitoring for its toxicities using digital technology have begun at certain centers. The groundwork for offering the outpatient infusion of CAR-T cells and outpatient monitoring of CAR-T patients predates the COVID-19 pandemic. At the 2019 American Society of Hematology Annual Meeting, Bachier, C. et al. presented data on outcomes following the outpatient administration of liso-cel in three ongoing clinical studies for relapse/refractory B-cell Non-Hodgkins Lymphoma [48]. Patients were required to have a caregiver for 30 days post-liso-cel infusion, receive safety-monitoring education (recognizing fevers and other adverse events), and to stay within 1 hour from the site of care, the university center. There were 37 patients, including patients above the age of 65 years and with a high tumor burden who received liso-cel infusion in an outpatient center. Of the 37 patients, 22 (59%) required hospitalization, 1 patient required ICU-level care, and 15 (41%) patients were followed in the outpatient setting [48]. The low incidence and late onset of CRS and ICANs allowed for outpatient administration of liso-cel and created opportunities to expand on this outpatient platform [49]. Risk assessment strategies are being evaluated to identify patients for outpatient vs. inpatient monitoring, using LDH, CRP, Ferritin, metabolic tumor volume, performance status, and the presence of CNS disease or multiple co-morbidities [50]. The time to CRS and ICANs varies for each construct, and the center’s readiness to admit and urgently treat patients will also determine if the infusion will be given in the outpatient setting. Reimbursements for CAR-T therapy do play a role in this decision-making and that is beyond the scope of this review.

The University of Oklahoma’s outpatient CAR-T program includes eight key components of the program with one called “Alert and Educate Community Providers”. As part of this outpatient program, community providers have access to a hotline number that connects them to a cell therapy charge nurse and on-call physician 24 h a day/7 days a week [16]. Similar outpatient programs exist at several centers across the US and allow the opportunity to offer this therapy in the outpatient setting, thereby leaving hospital beds available for patients requiring inpatient care. Similar to HCT patients, CAR-T patients are a high-risk population for infections, and it became a priority to minimize their COVID-19 exposure during the pandemic. In 2020, the CAR-T consortium investigators published practice guidelines on CAR-T cell treatment and toxicity management, particularly focusing on pre-CAR-T assessment and management of its low-grade complications [51]. The use of telemedicine was recommended for the initial consultation to determine eligibility and to avoid in-person exposure. Patients and caregivers were given a thermometer and other devices to take vitals which allowed for safe monitoring via telemedicine after day 7 of CAR-T infusion. The pandemic has likely accelerated the growth of outpatient programs for CAR-T cell therapy out of necessity. Cellular therapy centers have formalized outpatient programs to safely deliver CAR-T treatments, and some have incorporated telemedicine in their practice. 

### 8.2. Integration of Telemedicine in CAR-T Cell Therapy

The cellular therapy program at Vanderbilt University had an integrated telemedicine program for recipients of CAR-T treatments and provided close monitoring of patients by a nurse and an APP who held the CAR-T phone for all questions and emergencies [17,52]. They performed twice a day in-person visits and one video visit overnight for 14 days after CAR-T cell infusion. Patients and caregivers were trained to take vitals and evaluate CRS and ICANs at home. They had 13 patients who received outpatient axi-cel or brexu-cel followed by post-infusion monitoring with this telemedicine model. The median age of this cohort was 64 years, with six patients above the age of 65 years. The investigators reported that patients preferred outpatient monitoring via telemedicine, and it was feasible and a safe approach. Future interventions to improve outpatient treatment and care of CAR-T patients should include the incorporation of digital tools for toxicity monitoring, such as wearable health devices and electronic encephalopathy assessments [53]. A recent review on the implementation of digital health for CAR-T patients identified potential time points to incorporate vitals (via wearable devices) and electronic patient-reported outcomes to assess patients for CRS and ICANS [54]. Ongoing research using machine learning to determine optimal toxicity management and the development of clinical decision support systems will be valuable in delivering safe and efficient care using telemedicine for CAR-T patients.

There are several ongoing trials pursuing CAR-T infusion and post-infusion management in the outpatient setting using wearable devices. For example, a trial at Vanderbilt University is evaluating telemedicine to demonstrate the safe administration of axi-cel in the outpatient setting using wearable devices and frequent telemedicine visits (NCT05108805). Zuma-24 is evaluating the safety and efficacy of axi-cel and the concomitant use of prophylactic steroids in the outpatient setting for patients with relapsed/refractory large cell lymphoma (NCT05459571). The primary endpoint of this study is the incidence and severity of CRS and ICANs, and the study also aims to evaluate QoL and hospitalization in addition to disease and CAR-T-related outcomes. This is a multicenter trial that is currently recruiting at 10 centers in seven states that have an existing outpatient CAR-T program, and it is likely to shine light on regional/population-specific outcomes. While the use of telemedicine in CAR-T therapy is still in early implementation at many centers, we anticipate the increasing importance of this modality as the development of safer CAR-T products will increase accessibility to this therapy in an outpatient setting and potentially outside of a large academic cellular therapy center.

## 9. Conclusions

Delivering successful telemedicine care for HCT and CAR-T patients is complex, and it will require well-designed trials to identify an effective model of care for the target population. The available data suggest that telemedicine can enhance the current care models for HCT and CAR-T therapy, not only in the acute treatment phase, but also for long-term care. As alloHCT patients often develop at least one chronic medical condition after HCT, a model of care such as the chronic care model (CCM) may improve the overall health of these long-term survivors. The goal of incorporating telemedicine into our practice for HCT and CAR-T patients should be to minimize the healthcare-related burden for patients and caregivers. Furthermore, it should allow for more time at home, away from the hospital and clinics, while meeting the medical needs during different stages of treatment. Future studies with telemedicine will show if this improves healthcare delivery, long-term engagement with the HCT or CAR-T team, and early detection of complications for timely interventions despite their geographic location. This will ultimately improve disease and quality-of-life outcomes while potentially reducing healthcare utilization and cost. 

Participating in clinical trials can lead to an added burden on patients and caregivers, especially for trials focused on HCT and CAR-T therapies. The costs related to clinical trial participation, such as travel, lodging, and food, create significant challenges to patients despite some reimbursements provided by the trial sponsor. During PHE, we obtained informed consent for clinical trial enrollment via the telemedicine platform and conducted virtual visits for routine follow-up visits in order to minimize the patient’s exposure to COVID-19. These practice patterns are shining light on the fact that we require significant effort from our patients and their caregivers when we enroll them in clinical trials, resulting in significant trial-specific burdens. There are components of the clinical trial evaluations, such as history and limited examination, that may be performed via telemedicine, supplemented by lab testing conducted locally, as we did during PHE. We need well-designed clinical trials to successfully incorporate telemedicine in the care of HCT and CAR-T patients.

Time spent at home is a surrogate for QoL and the overall financial burden since oncologic care often requires a commitment from patients and caregivers. Measuring patient outcomes by time spent at home has been the focus of many novel oncologic therapies. We have been conducting HCT and CAR-T therapy in a hospital-based system; however, newer HCT strategies using novel antibody-based conditioning regimens or graft manipulation that requires less exposure to toxic GVHD prophylaxis drugs may allow successful outpatient treatments. The use of a hybrid platform that incorporates telemedicine or remote monitoring combined with in-person visits for outpatient HCT and CAR-T will likely increase access, reduce costs, and improve QoL for our patients.

## Figures and Tables

**Table 1 cancers-15-04108-t001:** Telemedicine in acute care setting for HCT and CAR-T patients.

Study	*n*	Study Type	Treatment	Telemedicine Modality	Comments/Outcomes
Duke (Sung, 2022) [10]	25	Prospective, case control study	autoHCT alloHCT	Homebound HCT. Home visits by advance practice providers and videoconference with the physician, synchronous	Safe and feasible. autoHCT and alloHCT patients had comparable outcomes to their matched cohorts. autoHCT patients had improved QoL, while alloHCT patients did not complete surveys for evaluation
MSKCC (Landau, 2022) [12]	15	Prospective	autoHCT	Homebound HCT. Telemedicine by advance practice provider, synchronous	Safe and feasible. Patients and caregivers reported preference. Technology was the barrier
SMARTCOVID19 (Mussetti, 2021) [13]	16	Prospective	alloHCT autoHCT	Real-time monitoring via mobile telehealth	SmartApp to detect abnormal vitals and symptoms
Duke-TRU-BMT (Racioppi, 2021) [14]	32	Surveys	alloHCT/ autoHCT	Mobile telehealth	Feasibility study
MSKCC (Nawas, 2020) [15]	20	Prospective	autoHCT alloHCT	Mobile telehealth cart, synchronous	Safe and feasible. Physician dissatisfaction was due to poor connectivity and lack of accurate examination
Oklahoma (Borogovac, 2022) [16]	21	Prospective	CAR-T	Outpatient infusion with a 24 h hotline and access to care team	Fifteen patients (71%) were admitted within 30 days, and the median number of days to admission was 4 (range 1–28). Majority for grade 1–2 toxicities. No deaths
Vanderbilt (Dholaria, 2022) [17]	13	Prospective	CAR-T	Combined outpatient clinic and telemedicine	Outpatient visits for lymphodepletion and twice-daily in-person visits post-CAR-T infusion until day 14. Combined with overnight remote visits with the help of telemedicine devices and round-the-clock access to a CAR-T provider

**Table 2 cancers-15-04108-t002:** Telemedicine in supportive care for HCT patients.

	*n*	Supportive Care	Treatment	Telemedicine Modality	Comments
Fred Hutch Cancer Research Center (FHCRC)	N/A (active program)	Survivorship/LTFU program	alloHCT	Telehealth via nursing and advance practice providers	Well-established program which connects patients and community physicians with a transplant specialist at the FHCRC
City of Hope (Chang, 2020) [26]	18	Prospective, early detection of cardiovascular complications	alloHCT autoHCT	Real-time monitoring	Patients and clinicians preferred telemonitoring for acute symptoms
Chronic GVHD (Turner, 2021) [27]	46	Prospective, early detection of Bronchiolitis Obliterans	alloHCT	Home spirometry telemonitoring	Feasibility and adherence. High correlation of home FEV1 with laboratory FEV1. In all, 9 of 12 patients diagnosed with BOS had FEV1 decline noted on home spirometry
Italian Report (Lupo-Stanghellini, 2020) [28]	236	HCT practice during PHE	alloHCT	N/A	Risk-stratified approach during the COVID-19 pandemic to determine if patients are appropriate for video vs. in-person visit
INSPIRE (Syrjala, 2018) [29,30]	>700	Survivorship, symptom management	alloHCT	Internet-based, interactive program	Randomized trial. Showed intervention arm had patient who reported “distressed” to “not distressed” status
City of Hope (Lee, 2023) [31]	20	Physical therapy	alloHCT autoHCT	Telehealth exercise intervention	Randomized trial (10 vs. 10 patients) over 8 weeks using resistance bands. Feasibility study. Showed improved gait speed and hand grip strength
Taiwan (Wu, 2020) [32]	14	Palliative care practice during PHE	Terminal cancer	Smartphone family conference	Variable responses from family members. Negative experience due to technical difficulties

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
