# Peer review of "Telemedicine in Hematopoietic Cell Transplantation and Chimeric Antigen Receptor-T Cell Therapy"

_cancers, 2023, doi:10.3390/cancers15164108_

Round 1
Reviewer 1 Report
The Gandhi & Lee article presents an overview of the current and potential role of telemedicine in hematopoietic cell transplantation and CAR-T cell therapy. While the study is well written and engaging, there are a few issues that could be improved:
1. The titles of the tables fail to provide a clear description of their contents. For instance, Table 1 is titled reads "Studies in acute care setting for HCT and CAR-T patients." It is important to explicitly state that these studies concern to telemedicine. By providing more descriptive titles, readers can quickly understand the focus of each table.
2. The conclusion should be rewritten to be more concise and focused. Rather than introducing new concepts or discussing specific programs like the Chronic Care Model and the SMILe-ICM program, it should solely focus on summarizing the conclusions derived from the available evidence. The conclusion should also highlight future considerations based on the article's findings without delving into program descriptions, which can be presented in separate sections within the article.
By addressing these aspects, the article can enhance its clarity and effectively convey its key findings on telemedicine's role in hematopoietic cell transplantation and CAR-T cell therapy.
Author Response
- Titles modified.
- Conclusion revised as suggested. Agreed and removed info on the SMILe program from the conclusion and added it to the body of the paper
Reviewer 2 Report
The review article titled Telemedicine in hematopoietic cell transplantation and chimeric antigen receptor-T cell therapy" by Gandhi and Lee examines a new and interesting topic in health care, namely Telemedicine. Although in its infancy, this is an interesting area.
The manuscript is well written and contains adequate number of relevant references. The authors have examined various (and relevant) important areas.
I recommend its acceptance in current form.
minor editing needed
Author Response
Revised grammar once more. thank you for the positive feedback.
Reviewer 3 Report
This is an interesting overview of the use of telemedicine in HCT and CAR T patients, an increasing use tool.
The main problem
-Too long manuscript, mainly narrative, visually not very attractive. Each section is a “block” of text that is not easy to follow. What is the message of each section?
It would be helpful to have a brief summary at the end of each section, and more tables with the important information/messages of the different topics reviewed across the manuscript.
For each section, it would be more informative to have a similar structure. For example: an initial summary of the main points or needs of the different patient populations or aspects under review; followed by an organised description of the different studies with telemedicine and their results; and a final summary of the situation of telemedicine in that section.
-Some points focused on the USA health care system, which has a limited extrapolation to other systems. This point should be commented on in the manuscript.
a) Abstract
-Is narrative but not very informative.
What is the evidence of the use of telemedicine in these patients? Do we have enough evidence of its utility? Do we have evidence of the impact on the care of these patients? Is it used in a few centers, or it has a wide application?
-Line 22-23.
When you say, “ As a result, it is an optimal time to review our experiences with existing healthcare models for 22 HCT and CAR-T treatments, …”, do you refer to the experiences of the 2 authors of the manuscript or the experience in general at different centres?
b) Telemedicine in the cancer care continuum
-The authors differentiated telehealth from telemedicine. Please, add the references that support these definitions, or are they authors' definitions?
-Line 64
The authors state that there are four main modalities of telemedicine. Is this a widely accepted classification, or is the authors' opinion?
-Lines 91-92
"They are typically required to relocate for 100 days with caregivers providing 24-hour support."
This is a frequent situation for patients in the USA but not in many European countries where the transplant is performed closer to the patient's home.
-Line 96-97
"Historically, telemedicine has seen minimal use in managing recipients of HCT or CAR-T therapy" Please, could you give references, and numbers to quantify this statement? Is there some quantitative information about the use of telemedicine historically and nowadays?
c) Tables
-Add the reference number to the different studies. For some of them, I can’t find the reference in the references section, or there is more than one for the same author.
-Table 2.
It is surprising that for the FHCRC no patient number is given, although in the abstract, the authors said that “ As a result, it is an optimal time to review our experiences …”
-As commented before, having more tables with the important information/messages of the different topics reviewed across the manuscript would be helpful. What are the pros and cons of telemedicine? Which patients or situations are more suitable for telemedicine, and which are, in principle, not well adapted for it?
d) Barriers with TeleMedicine
Some points apply to the American health system, but not to many of the European countries' systems.
For example, the different types of reimbursement of telephone visits, compared to video and in-person visits, do not occur in the public European Health Systems. This is not an issue in these systems to disincentivize physicians and institutions from providing care via telemedicine.
e) Telemedicine in Chimeric Antigen Receptor – T (CAR-T) cell therapies
Again, it is focused on the USA Health system. The FDA approvals have no application to other countries with other regulatory agencies (EMA, for example).
I think it is unnecessary to list the different approved CAR T products.
f) -Line 99. Reference 24 seems wrong here. I understand that it should be reference nº 27
g) Conclusions
It is too long with comments that should be in other sections.
-For example, the detailed description of the CCM model or the SMILe-ICM program is not for the conclusion. You can comment here about this model or program, but it is not the place for its detailed description. The same is true for the NCT04789863 study.
-Clinical trials have not been mentioned before the conclusion section. Again, here you can make a comment on this topic, but it is not the place for an extended text setting the situation, the problems of cost, the burden for patients and caregivers, aspects that can be done with telemedicine instead of face-to-face visits, …
Author Response
See attached. Responses in Red

Round 2
Reviewer 3 Report
The authors have improved the manuscript, introducing some subtitles, moving part of the text of the discussion to another section, adding references to the tables and shorting a little bit the manuscript. The manuscript continues to be dense. Each section is a “block” of text that is not easy to follow. What is the message of each section? This remains the same.
The focus of the manuscript is the American health system, without acknowledging that there are different systems The different types of reimbursement of telephone visits compared to video and in-person visits, sure that is a big issue in the US but sure not in other health systems. As I suppose that there are some/quite readers outside the US, a wider view will be welcome.
The users of telemedicine are physicians, haematologists in this case, that are not necessarily experts in the field of telemedicine. In consequence, explaining basic concepts will be welcome. The majority of readers are not experts in telemedicine but experts in the transplant and CAR T fields. A more educational view will facilitate the reading and understanding of the manuscript. Even if widely accepted or known by the experts in telemedicine, adding references to support the differences between telehealth from telemedicine and the four main modalities of telemedicine will facilitate the reading and understanding.
Author Response
Thank you for your feedback. Your comments and suggestions have been helpful. I think you will enjoy reading the revised version. Please see responses below.
Response to comment 1: We have shortened text and organized sections based on your suggestions.
Response to comment 2: Statement modified to suggest there are differences in reimbursement system.
Response to comment 3: reference added
Thank you and we look forward to hearing back from you.
